# IIoT System for Intelligent Detection of Bottleneck in Manufacturing Lines

Manuel José Rodríguez Aguilar *, Ismael Abad Cardiel and José Antonio Cerrada Somolinos

Department of Computer Science and Automatic Control, School of Computer Science, Universidad Nacional de Educación a Distancia (UNED), Juan de Rosal 16, 28040 Madrid, Spain; iabad@issi.uned.es (I.A.C.); jcerrada@issi.uned.es (J.A.C.S.)
* Correspondence: mrodrigue6814@alumno.uned.es

**Abstract:** Production lines face numerous challenges to meet market demands, including constant changes in products that require continuous adjustments. Efficient and rapid reconfiguration and adaptation of production processes are crucial. In cases of inadequate adaptation, bottlenecks can arise due to human errors or incorrect configurations, often introducing complexity in pinpointing the root cause and resulting in financial losses. Furthermore, improper machine maintenance contributes to this situation as well. This article seeks to establish a framework grounded in the contemporary smart factory, the IIoT, the Industry 4.0 paradigm, and Big Data. The proposed system places emphasis on leveraging real-time data analysis for predicting risks, while concurrently conducting a thorough analysis of historical data to monitor trends and enhance bottleneck identification. The defined architecture operates across multiple levels, acquiring real-time information and generating historical data for training and continuous optimization. Predictive results contribute to decision-making and assist in mitigating bottlenecks in manufacturing lines.

**Keywords:** bottleneck; manufacturing line; Industry 4.0; Industrial IoT; Big Data; Machine Learning

## 1. Introduction

A bottleneck is a process, machine, or productive stage of the system that impedes throughput from working effectively [1]. The reason for this situation is inadequate work, or having a low level of productivity. Consequently, it causes a delay within manufacturing operations, in addition to impacting the rest of the stages of the manufacturing line. There are different causes of the bottleneck; for example, inefficient maintenance on the machines, constant stops, non-adaptation to the demands of the rest of the line, slowdown of the supply chain of the manufacturing line, and untrained production and maintenance personnel not knowing how to correctly interpret information and incorrectly managing workstations. Furthermore, the matter is not always visible. Hence, a crucial aspect lies in the smart handling of data within the manufacturing context.

Thanks to technological advances in industrial operational technologies (OT) in Industry 4.0 and the Industrial Internet of Things (IIoT) [2], processes for acquiring information from sensors and actuators have been improved. Advances in communications and wireless technologies, such as 5G, can capture information quickly. Aspects such as Big Data and data analytics have also contributed to classifying product line problems to compare and prevent different situations.

After a literature review on these aspects, we found proposals [3] such as data analysis throughout diverse types of uncertainty, such as market variability, execution of indistinct services, or the user. Other authors [4] proposed the cognitive enhancement aspects or Machine Learning to improve safety and industrial maintenance activities. Regarding the bottleneck in assembly lines, the authors of [5] directly dealt with its resolution through the technical application of line balance using the lean manufacturing methodology combined with the IoT.

Moreover, studies such as [6–8] delve into the application of advanced systems in industrial maintenance. In these studies, the authors investigated, in a general way, crucial factors, such as the captured data quality, and logical factors, such as the existence of a uniform information system between stations and human conditions, maintenance, and the performance of machines. Augmented reality is another technological advancement. The authors in [9] specified an approach to addressing the bottleneck on production lines based on cycle time calculations collected using product-specific RFID tags to identify bottlenecks.

In conclusion, concerning the studies investigated, most of them agree on three fundamental points to consider related to optimization and maintenance aspects: the data quality, the information consistency in the manufacturing process, and the data acquired from sensors and actuators. As we have seen, there are numerous studies about the bottleneck issue, mainly focused on analysis and simulation. The reason for this study and the gap we observed with respect to previous studies is the need to adjust this analysis to a current smart factory paradigm based on current Industry 4.0 references. The objective is to establish an advanced method to detect bottlenecks in manufacturing lines. Specifically, to create a framework to predict bottlenecks from data analysis. Therefore, this work complements the previous analysis of this issue with a proposed architecture based on current IIoT models and communication standards to create a system for detecting bottlenecks in manufacturing lines.

The system relies on three core components: acquiring and analyzing key data [10], primarily from production line sensors and actuators, processing and integrating data using an Industry 4.0 reference architecture [11], and training these data through a Big Data system for bottleneck prediction.

The rest of the article organizes the content as follows: Section 2 is a chapter dedicated to the literature review. The next chapter, Section 3, outlines the core proposal of the article, presenting a detailed description of the system designed to detect bottlenecks in production lines. Section 4 shows the experimentation of the proposed model. It includes a model workstation with sensors that acquire measurement simulation, with the different technologies defined at each level. To showcase the solution, the IIoT software v1 architecture is implemented within a server environment, enabling the acquisition and processing of information based on the implemented logic for bottleneck detection in production lines. This section also presents the different metrics designed, the measurements sampled, and the results obtained. Finally, we present a discussion of the system, as well as a section to define challenges, open issues, future work, and conclusions.

## 2. Literature Review

First, we conducted a brief review of the bottleneck problem in manufacturing lines and the commonalities in the literature. Second, we reviewed the technological advances that directly impact the information flows necessary to know the status of industrial sensors and actuators and integrate and manage quality data in a smart manufacturing ecosystem. Third, we reviewed the "Big Data" concept and how it impacts manufacturing environments. Finally, we conducted a review of reference architectures based on Industry 4.0.

### 2.1. The Bottleneck in Manufacturing Lines

In the reviewed literature [1,9,12], terms related to the bottleneck in production lines included congestion, capacity below demand, process limiting throughput, and production blockage. Essentially, a bottleneck signifies an obstruction to the regular flow of production. These situations frequently occur on manufacturing lines due to various factors, such as improper synchronization between workstations, poorly formed products hindering proper flow, or inadequate maintenance of a machine, leading to operation below its normal performance level.

In production lines, instances often arise where machines or workstations exhibit asymmetry in the product processing speed. To address this, buffers or accumulators are installed to equalize the overall capacity of the manufacturing line. However, mis-

calculations in buffer size, in response to production demand, can result in bottlenecks by creating product accumulations that disrupt workstation performance. Additionally, situations involving product dependencies may occur, such as the absence of an additive hindering proper mixing in paint manufacturing or the lack of flavor aroma affecting the filling process of yogurt.

Additional causes include inefficient maintenance of machines, frequent stoppages, failure to adapt to the demands of the rest of the line, slowdowns in the manufacturing line supply chain, and inadequately trained production personnel.

Once the bottleneck arises, a thorough investigation and identification of its origin become necessary. This involves a comprehensive examination of the production line, reviewing production indicators and external dependencies. Monitoring critical sensors and actuators, including examples such as counting detectors, vibration sensors, or motor noise detectors, becomes essential to detect faults in machine components. Although they may not be within the control range of the machine or workstation, keeping sensor and actuator values within acceptable limits is crucial to avoiding product failures or problems with machine components. Therefore, analyzing these sensors and actuators will be vital to detect future bottlenecks through predictive analysis. Current technologies based on IIoT and Industry 4.0 standards facilitate real-time capture of these components and sensors, enabling efficient data generation and training for bottleneck prediction.

Technological advancements and enhanced production data analysis have had a significant impact on manufacturing operations, enabling improved predictions of maintenance needs and failure forecasting [6–8]. Initiatives, as explored in studies such as [9], leverage machine vision and simulation to enhance the identification of bottlenecks in the manufacturing process.

The authors uniformly underscored three crucial elements in their analysis: data quality, consistency in production information, and data from sensors and actuators. This work puts forth a framework focused on these aspects to capture and integrate key data, with the objective of improving bottleneck identification. For an in-depth exploration of the proposal components, please refer to Section 3.

### 2.2. Technological Advances That Impact Data Flow and Data Integration

Technological advancements in networking and industrial protocols are revolutionizing traditional control systems, enabling flexible information flow and communication within and between industrial layers. Notably, the optimization of industrial Ethernet protocols, such as Time-Sensitive Networking (TSN) with features such as Single-Pair Ethernet (SPE), consolidates diverse sensor and actuator data into a unified network. This not only reduces wiring but also supports data-driven manufacturing by facilitating increased sensor and I/O device installations, as highlighted in [13].

The growing adoption of 5G technology in industrial communications is enhancing the bandwidth capacity, reducing latency, and improving the overall efficiency. This technology, standardized for smart manufacturing applications, supports the Industrial Internet of Things (IIoT) and industrial wireless communications [14,15]. The integration of Ethernet TSN and 5G offers increased potential for automation, sensor augmentation, and smart manufacturing. Additionally, the rise of devices with 5G interfaces proves advantageous in situations where cable usage is impractical, such as in hostile environments with chemical agents.

The 2023 Industrial Network Market Share, as reported in [16], highlights substantial growth in industrial Ethernet, surpassing wireless technologies. Although traditional fieldbus is decreasing, it remains prevalent in industrial control. The study indicated a 68% market share and 10% annual growth for industrial Ethernet-related network technologies, while wireless technologies hold an 8% market share with an annual growth of 22%. In contrast, fieldbus accounts for 24%, experiencing a negative annual growth of −5%. These data reflect a shifting preference toward wireless and Ethernet networking for interconnecting field components.

With the data acquired from sensors and actuators, advancements in network standards support both vertical and horizontal information flow, aligning with the Industry 4.0 paradigm [17]. Technologies such as MQTT [18] enhance message exchange, facilitating rapid data delivery to other software layers or the cloud. Protocols such as HTTPS and Modbus TCP improve communication latency over the industrial Ethernet network, enabling smoother interaction between devices, such as programable logical controller (PLC) and other clients, such as those using Web or Modbus, for information retrieval.

The OPC protocol with UA mode [19] presents a service-oriented architecture that integrates components, incorporating features such as security and modeled information. This protocol facilitates the exchange of information with programmable controllers, middleware systems, and integrated cloud-based systems [20].

In summary, the ongoing changes in technology are transitioning from a monolithic focus on individual machine control to a flexible, integrative, and scalable approach. This shift brings forth various features and capabilities, as follows:

- Establishes information connections between processes, workstations, factories, and remote sites.
- Makes this information flow more accessible and expands the scope to analyze issues, such as those in this article, to obtain an effective detection of bottlenecks.
- Provides higher-quality data [21], encouraging monitoring, measurement, and comparison with other key information, where we will be able to control deviations or trends.

The 6G wireless technology, discussed in [22], promises future industrial scenarios based on the Internet of Verticals, emphasizing cloud manufacturing and the provision of resources as services. This approach allows for the sharing of solutions between workstations or similar machines. However, a significant challenge is ensuring the security of information flows due to increased mobility, leading to more frequent cyber-attacks. In [23], the authors underscored the importance of classifying cyber-attacks by layers in IIoT architectures. Moreover, the authors of [24] outlined different vulnerabilities in the current industrial control context.

### 2.3. Big Data

Big Data plays a crucial role in the fourth industrial generation, as highlighted in various studies. In [17], its influence spans areas such as improving fault tolerance and predicting machine states. The authors of [11] emphasized the significance of Big Data in rapid failure detection for smart manufacturing, essential for enhancing industrial maintenance practices. The authors also noted its importance in additive manufacturing with 3D-printing technologies [25] and machine design. Moreover, the inclusion of bottleneck detection is considered as part of its broader application in critical analysis.

In [10], the author explored challenges in information processing, focusing on computational complexity and the difficulty of extracting valuable insights from vast amounts of raw data. The discussion included challenges in monitoring the health status of factory machines using cloud computing and the need for high-performance local machines for effective local computing. The development of new distributed systems for data analysis is considered a challenge, requiring substantial computing power. Additionally, there is a debate on whether processing such massive amounts of data is necessary, with some authors emphasizing the importance of maintaining quality information over quantity.

In [11], the researchers outlined key analytical methods for industrial Big Data. Descriptive analytics involves interpreting historical business-oriented information, while predictive analytics enables the detection of future events based on historical data, particularly useful in applications such as predictive maintenance. When combined with statistical models and AI, predictive analytics allows for predicting machine trends and behavior. Lastly, prescriptive analysis contributes to better decision-making, closely linked with predictive analysis to define actions based on prediction results.

The conclusions we drew from the authors in relation to the context of this work are:

- It is necessary to select which data to use to balance the cost and complexity of processing large amounts of information.
- Data quality: use cases where bottlenecks are reproduced should be captured and a specific dataset with key variables should be generated for analysis.
- Data analysis: predictive analysis has been proven by the authors to be the most effective way to analyze historical information and predict future events.

The Figure 1 displays a series of sets with nodes that act as data sources. The concept it represents is the grouping of these data sources into subsets that are crucial for specific analyses, such as aiding in the identification of bottlenecks in production lines.

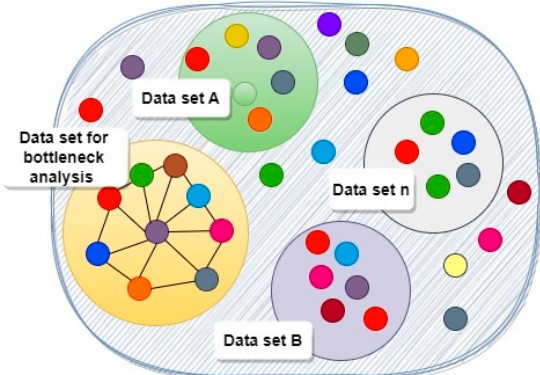

**Figure 1.** Example of grouping by dataset. Nodes represent data sources such as sensors, actuators, machine data, production, yields, etc.

Supervised SVM Algorithm to Build a Prediction Model in Big Data

In the realm of Machine Learning [26], supervised learning algorithms, including Support Vector Machine (SVM), Random Forest (RF), and k-Nearest Neighbors (KNN), are utilized for developing predictive models based on input and output data, particularly in the context of predictive maintenance, according to [27]. Additionally, the authors of [28] demonstrated the use of SVM classification for weld quality prediction and its potential in real-time monitoring systems, emphasizing the importance of an appropriate feature selection method for achieving higher accuracy.

Therefore, since our system monitors sensors in real time, we will leverage this synergy to utilize the algorithm. By adjusting it, we will use it for data training, consequently making future predictions on sensitive datasets for bottleneck detection. On this same SVM algorithm, the authors of [29] cited it as recursive for predicting a relationship between product shape elements and product images. Although these authors likewise considered the k-Nearest Neighbors (KNN) algorithm in their analysis. This algorithm will also be used to make comparisons with SVM and will have an open-contrast base.

### 2.4. Current Reference Architecture for Industry 4.0

Two primary architecture models were identified within the industrial context of manufacturing. One of them was exposed by the IIC (Industrial Internet Consortium) in 2017 as a common framework, called the IIRA [30] (Industrial Internet Reference Architecture). This architecture defines a high-level reference table with models, sectors, processes, and a series of domains to create an entire base ecosystem. Structurally, this IIRA defines a system through tiers that allow the information technologies model to integrate with the operational technologies model. It includes hardware and software in the physical context of industrial manufacturing, such as PLCs or SCADA systems [31].

This architecture, illustrated by the IIC [30], was partially founded on the principles of edge computing [32].

As we can see in Figure 2, the model comprises three tiers: edge, platform, and enterprise. The edge tier acts as the integration point between the industrial environment

and information technology, connecting control devices, regulation devices, and sensors. The platform tier, facilitated by the edge gateway, includes components for information transformation, functional scope implementation, and data analytics. This tier establishes a data flow for integration with the enterprise tier, where applications develop business logic, rules, and controls.

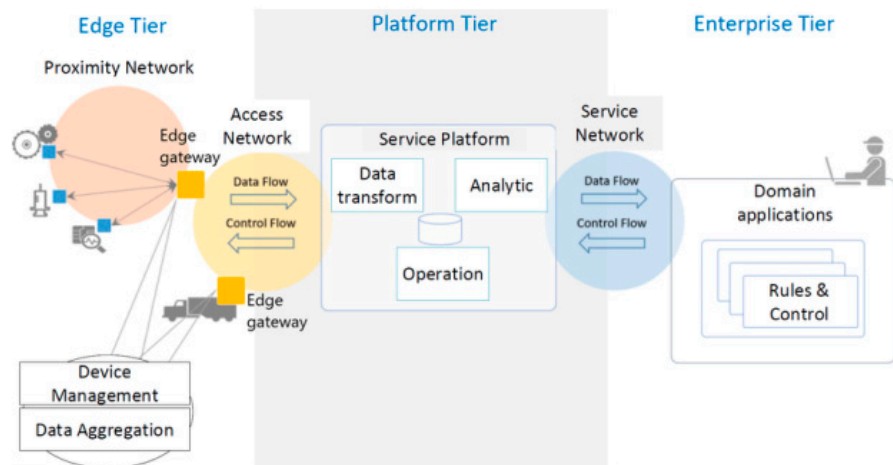

**Figure 2.** IIRA reference: three-tier IIoT architecture (Reprinted with permission from [30]. Copyright 2023 Object Management Group).

The key to the model's effectiveness lies in the seamless interaction and communication of data between tiers. Continuous traceability in data communication is highlighted, particularly in real-time scenarios. Additionally, timely acquisition of sensor data is emphasized, as demonstrated in addressing bottlenecks in a production line within the article.

RAMI (Reference Architectural Model for Industry 4.0) is the other model proposed internationally by the body Platform Industry 4.0 [33]. As we see in Figure 3, this model proposes a vertical layer stack.

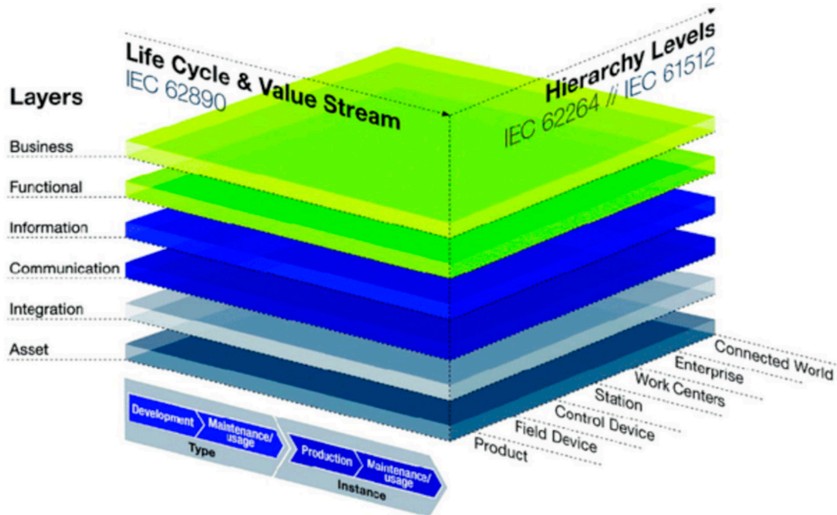

**Figure 3.** RAMI reference (reprinted from [34]).

The authors have developed a three-dimensional base model to facilitate Industry 4.0 discussions. This model adopts a service-oriented architecture, consolidating all elements and components of information technologies into a unified layer. As illustrated in Figure 4, the hierarchical structure depicts communication among various industrial elements and devices [35]. A key feature is the multiple interconnections among different components,

emphasizing the importance of considering various protocols for effective linking of diverse devices and components.

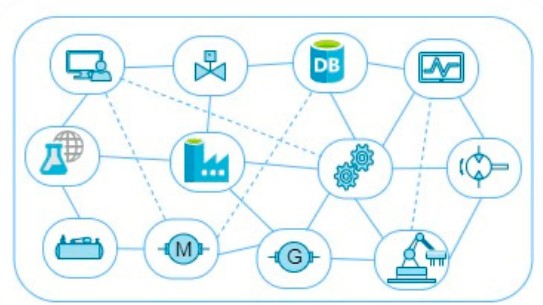

**Figure 4.** Interrelation of components of different types within the Industry 4.0 ecosystem.

Comparing the two models, both present division by layers, but the main difference is that the IIRA model presents an alignment between domains and cross-functions, while the RAMI model presents an interconnected system between all its components.

Another differential example is assets. The RAMI model sees an asset as any element that can be interrelated, such as a product, raw material, software, or people. In contrast, for the IIRA model, assets are the physical components that are controlled [36,37]. Given this service architecture, the RAMI model, unlike the IIRA, must have a powerful system of services.

In the literature, numerous articles incorporate both Industry 4.0 references. For example, in [38], the authors developed a forensic model for investigating industrial automation environments, utilizing the IIRA benchmark to create an architecture that enables internet access for forensic analysis against remote attacks on industrial control components, such as PLC and SCADA. In [39], the authors proposed a RAMI-based architecture with an additional layer, emphasizing an intelligent federation of components and human-centered intelligent decision support. Various manuscripts have explored future trends and alignment with standards, such as [40,41], where the authors reviewed and aligned both IIRA and RAMI references with the IEEE 2660.1 standard for industrial agents.

The proposed Industrial Internet of Things (IIoT) architecture follows the structure of the Industrial Internet Reference Architecture (IIRA). It is designed to be modular and tiered, aligning with the framework provided by the IIRA reference. This approach differs from the interrelationship model proposed by the Reference Architectural Model for Industry 4.0 (RAMI).

## 3. Proposal

The system we propose is based on three fundamental aspects:

- The first aspect is to acquire and analyze quality data from sensors and actuators of the manufacturing lines that provide us with relevant information about bottleneck risk. With this, we can create a dataset that will help us evaluate the situation and value the system.
- The second aspect is an Industrial IoT architecture that transforms and articulates this information in an agile way from the operational technology (OT) [42] environment to information technology (IT) [43]. From this point, we will obtain a classified and plotted structure of the acquired data in near real time to make decisions in the face of the problem.
- The third aspect is the Big Data system and historical data analytics that will provide a basis to compare with the quality dataset previously acquired. We have the capability to handle trained data frames, allowing real-time predictions based on data acquired at the field device level.

### 3.1. Quality Data

The proposed Industrial IoT architecture represented in Figure 5, focuses on the use of discrete and continuous values, prevalent in industrial control. Digital sensors generate binary signals, while analog sensors measure within a numerical range. These sensors and actuators can be connected to PLC interfaces, industrial PCs, or operate independently via network cables or wireless connections. In addition to sensor and actuator data, manufacturing information, such as production volumes, line performance, and downtime calculations, is crucial and is provided by control systems, such as PLCs and SCADA. External data, to complete the product, may also be essential for certain processes.

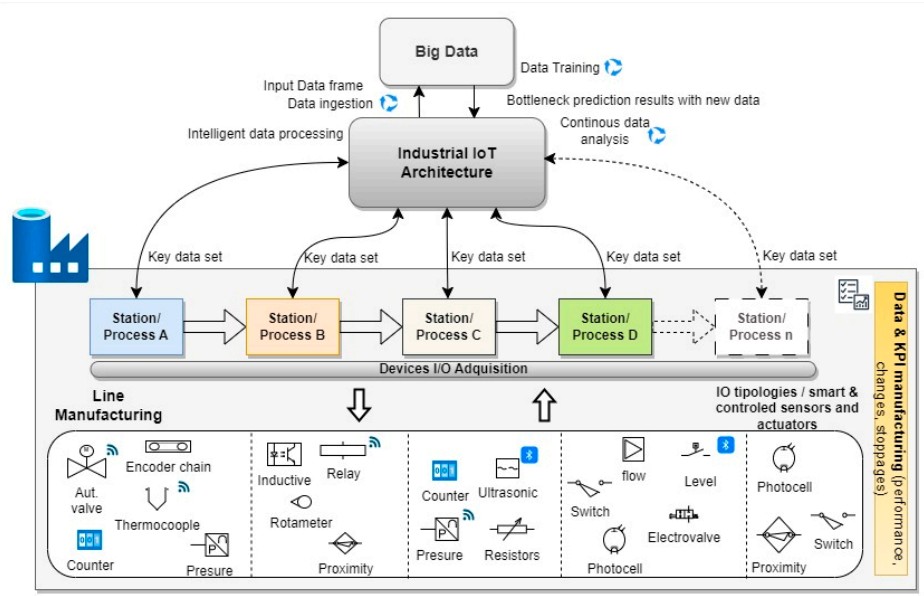

**Figure 5.** Proposed system for intelligent bottleneck detection in the manufacturing line.

In selecting pertinent data for analysis, one must consider the following aspects:

- The user experience of the workstations, processes, or machines of the manufacturing line will be important to consider when obtaining the dataset.
- The approach to philosophies, such as total productive maintenance (TPM) [44], is important to create multidisciplinary teams that acquire the knowledge about the manufacturing line or process to track and establish criteria to determine what data and signals from sensors and actuators are needed to define the information set related to the bottleneck.
- The support provided by a Big Data system or data mining of manufacturing variables in contrast to production results will also be strategic to determine which sensors have provided sensitive data in the face of yield decreases, shutdowns, delays, and other aspects related to the occurrence of bottlenecks.

### 3.2. Proposed IIoT Architecture

We propose an IIoT system based on Industry 4.0, specifically in the IIC reference, where we integrate operational and information technologies to generate a dataset that provides information for bottleneck detection. This is structured in the following tiers, shown in Figure 6.

In Appendix A, we present a basic flow of data preprocessing. In the subsequent lines, we present the definition of the various tiers within the proposed model:

- The field devices tier facilitates connections with control devices such as PLCs, micro-controllers, industrial PCs, embedded systems, and smart sensors. This layer features a communication interface emphasizing interoperability through various protocols,

including Ethernet and Wireless. While adapting this technology to an Industry 4.0 system may require effort, optimizing IT–OT convergence is crucial, as highlighted in [45]. Overall, this tier serves as a key link, connecting with control devices to acquire data from sensors and actuators.

- The edge tier plays a crucial role in processing acquired data for bottleneck detection. Using primarily MQTT 5 and OPC-UA [46] protocols, it prepares and categorizes data, with additional connectivity options, such as HTTP for external data sources. The integration component connects to these protocols, collecting sensor values and key data related to bottleneck issues. The objective is to subscribe to MQTT broker "topics" [47] or obtain data from OPC-UA clients, integrating, classifying, and publishing datasets for further analysis in the subsequent tier.

- The platform tier consists of a distributed event streaming platform and a microservices backend [48]. The event streaming platform subscribes to datasets containing bottleneck information, allowing real-time or deferred data utilization. The backend, operating as microservices, analyzes and publishes results by comparing real-time datasets with trained data from Big Data for risk predictions. Traceability data are stored in a relational SQL database for business use, including dashboard displays of data trends. Simultaneously, a NoSQL database stores data in JSON format, facilitating Big Data analysis and supporting data traceability analysis in the enterprise tier.

- The enterprise tier is designed to fulfill the needs of bottleneck detection and facilitate decision-making regarding risk. It includes a graphical system interface resembling an Andon system [49,50] for displaying bottleneck risks at key points on the manufacturing line. The front-end web client receives risk data from microservices, derived from Big Data information. Additionally, the component focuses on data analysis, observation, and monitoring over time using a NoSQL database type. This analysis aims to assess trends, calculate data, and create comparisons during moments of bottleneck risk, providing a solid foundation for informed manufacturing decisions.

- Data traceability is a cross-component application ensuring data quality by incorporating timestamps in each tier. This enables comprehensive tracking of information at every step, offering metrics for optimal path determination, especially across different network segments. Continuous measurement of data is conducted to verify their quality and detect deviations, with the enterprise layer utilizing this information for latency analysis.

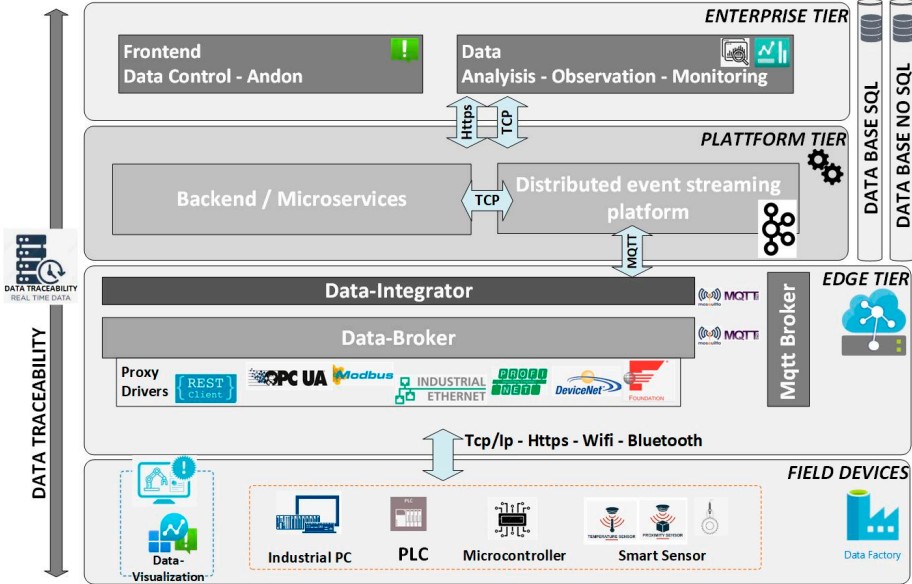

**Figure 6.** Proposed Industrial IoT Architecture Model.

### 3.3. Big Data and Machine Learning

In our investigation of identifying bottlenecks in the production line, we need a Big Data system with real-time processing capacity, considering that it will be carried out on a specific set of data that provide us with the information to be able to predict bottlenecks in the manufacturing line and to make the most of this work.

In the following sections, we will see the context of the Big Data system and how we fit the part of the input, data source, and its ingestion. We will also look at the other functions of the Big Data system for bottleneck detection.

### 3.3.1. Big Data Process

Within the logical context of a Big Data ecosystem, we have the following components, outlined below.

In Figure 7, the first component involves structured and unstructured data, including information from sensors, actuators, and key production data. In our proposed IoT architecture, these data reside at the platform level, integrated in real time for ingestion into the Big Data system. Detections during architecture execution are stored in the NoSQL database of the proposed architecture, serving as the foundation for creating a historical dataset for training predictions. The event streaming platform in the proposed IIoT architecture then publishes the trained dataset for the Big Data system, enabling real-time predictions.

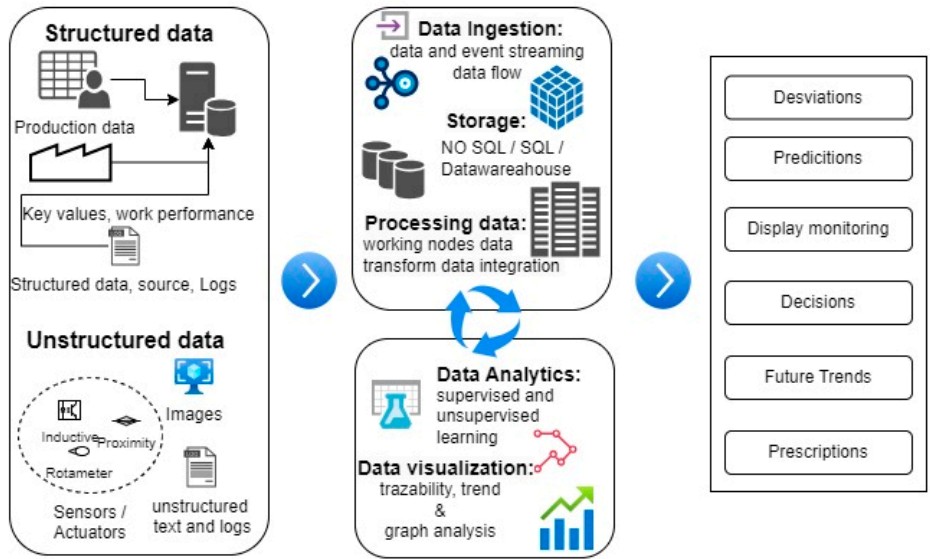

**Figure 7.** Information flow between Big Data system blocks.

The proposal limits the dataset to address bottleneck issues within a specific scope of the system. However, a key characteristic of Big Data is its scalability, allowing it to expand to other cases for similar detection and be shared. Information sharing, as suggested by the authors in [51], is deemed beneficial for product manufacturing systems, serving as a resource for various purposes.

The Spark architecture uses the streaming library to drive the dataset as events and can be run on an algorithm through the Machine Learning libraries [52]. This comparison will give us a result based on the trained information over the historical data. It is also necessary to have a larger storage system that will be driven to the HDFS platform of Hadoop [53].

### 3.3.2. Data Frame

To ensure a robust data frame for effective detection, we need data acquired from input and output devices susceptible to information related to bottleneck detection, key manufacturing values, and specific information about the source element:

- From sensors and actuators, we can have discrete data (all/nothing), accumulated information, and numerical values from analogue sensors, such as temperatures, humidity, flow rates, levels, etc.
- The key manufacturing data [54] are those that will give us information about the moment of manufacture, such as yields, number of machine stoppages in each time, number of format changes in each time, product manufactured, etc.
- Specific information of the origin element, such as identification of the process, line, machine, specific station, etc.

To build the prediction model using classification algorithms, it is essential to define the input and output data for the data frame. The target output should be determined by indicators and situations providing information about the bottleneck. Previous studies, such as [1,12,55], highlight various bottleneck indicators, such as the smallest isolation production rate (PR), work-in-process inventory (WIP), machine-specific performance, or line performance variability over time. For instance, if the bottleneck is associated with fluctuations in line performance, buffering-induced stops, and a critical temperature for the product, these factors would serve as output indicators in the model.

### 3.3.3. Interpretability and Explainability

Machine Learning's crucial aspects are explainability and interpretability. The authors of [56] explored various sources offering diverse interpretations of these terms. Both emphasize human comprehension of decision causes. Explainability relates to the clarity of internal algorithmic procedures during training or predictions. Despite numerous interpretations, there is no precise mathematical definition for these terms. The proposal emphasizes defining and knowing key production line variables crucial for bottleneck detection. Processed and integrated within the IIoT system, these variables contribute to generating a data frame with an output class containing essential bottleneck indicators, serving for both training and future predictions. The paper investigates various classification models in Section 2, determining SVM and KNN as the most suitable for predicting problems in real-time monitoring and industrial maintenance within the context of production machinery. Interpretability hinges on the model's depth: KNN explains output based on neighbors [29], while SVM's interpretability varies with the kernel used. Complexity is contingent on the volume of the input element set and the results in classifying the output.

To facilitate and especially to reinforce their interpretability and understanding, once they have been trained, or at the time of prediction with new data, we must have a knowledge checklist, which collects the inputs, or even a plan with the elements that are being measured, as shown in the following example in Figure 8.

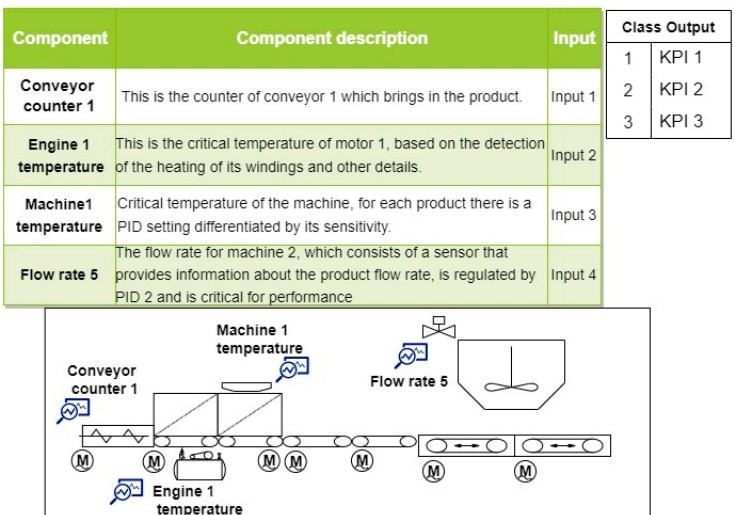

**Figure 8.** Example with a drawing of sensors and actuators for measurement and a table of details.

This will help when interpreting the result of a prediction and the data to be trained.

## 4. Experimentation

The case study involved a proof-of-concept to assess the performance of the proposed IIoT architecture and Big Data Machine Learning component. Two manufacturing machines controlled by PLCs were simulated, extracting key variables, such as temperature and motor turn counting, for bottleneck detection. Simulated changes within a measured range indicate bottleneck risk. The study began with data acquisition from a temperature sensor and lap counting at workstations.

The distinct tiers of the architecture will proceed with their workflow from bottleneck detection in the PLC devices to the enterprise platform layer, where the subsequent actions related to bottleneck identification will be undertaken:

- First, the Andon system [49,50,57] warns when a real risk of bottleneck occurs with the response provided by the Big Data system algorithm with the data trained through the input dataset provided.
- Second, the real-time graphical observation system captures the information and displays it on the timeline to analyze moments of risk. We also measured the historical trend of the variables considering when they impacted during the bottleneck risk. This allowed us to see if there was oscillation, focus the problem on the variable in question, and make decisions.
- Third, this experimentation environment also allowed us to cross between sensor and actuator measurements. In this way, we can compare their behavior, for example, whether key or risky moments aligned or not. Similarly, we measured the response times to make contrasts and comparisons of the system against latencies.

The experimentation offered insights into the interaction of technologies and communication protocols within the proposed architecture, focusing on layer integration and latency times tracked by timestamps in the traceability component.

The diagram depicted in Figure 9 shows the architecture at the system level with the interrelationship between its components; in the yellow legend, we can see the tier to which they belong.

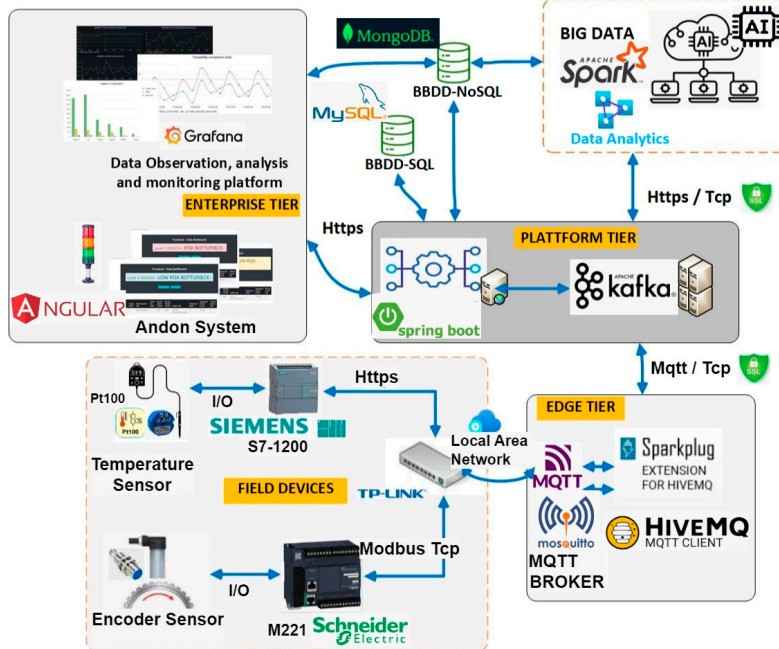

**Figure 9.** Systems diagram of the implemented experimentation.

### 4.1. Hardware Used

For the PoC, we used a Fischertechnik simulation based on the Industry 4.0 24V training factory model [58,59], which emulates a die-cutter and a synchronized conveyor belt with a temperature probe and two motors. The benefit of employing this model is that we can implement PLCs with identical control variables deployed in control and regulation for actual manufacturing stations, utilizing the same magnitude values.

The simulation to verify the proof-of-concept's operation in the proposed experimentation involved launching the components of the architecture on a local server. The simulation included PT-100 probe data connected to a prototype, where the machine temperature sensor was linked to an analog input in a Siemens S7-1200 PLC. Additionally, the prototype simulated a motor-turn counter connected to the digital input rack of a Schneider M221 PLC (See Figure 10).

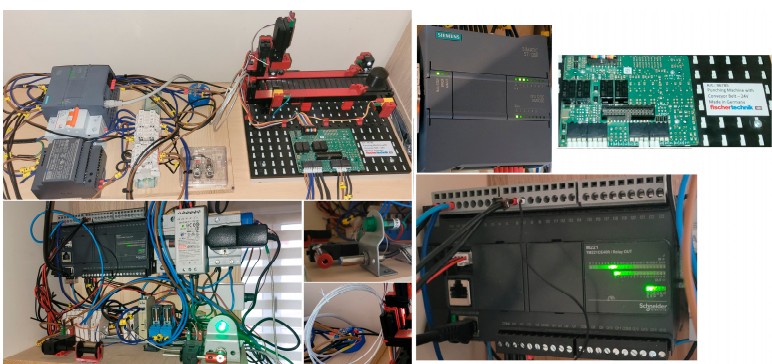

**Figure 10.** Detail of the Fischertechnik model and control electronics, with PLC Siemens S7-1200 and PLC Schneider M221, used for experimentation.

### 4.2. Technologies Implemented in the Proposed IIoT Software Architecture

Field devices and the edge tier:

- In the simulation implementation, at the edge tier, the Plc-Data-broker component publishes and subscribes the information acquired from the PLCs through the MQTT engine. We used a Modbus service for the Schneider PLC and a web service using REST protocol for the Siemens PLC.
- The sparkplug-data-integrator component follows the specifications of the sparkplug framework [60], integrating MQTT data with specific topics for dataset identification and classification at the edge level. Additionally, the system incorporates a quality level using the "data traceability" component to save timestamps linked to the integrated data.

At the platform tier shown in Figure 11, we will subscribe to the data published in the sparkplug-data-integrator to obtain a dataset with the information from the two sensors and their traceability. We will cover three objectives:

- The first is the distribution and persistence of data integrated by the event streaming component in Apache Kafka [61,62], arranged for the backend and Big Data.
- Second is the backend/microservices component interfaces with distributed real-time data, connecting with Big Data to utilize trained Machine Learning data. It sends new datasets to assess the risk of a bottleneck, with a microservice publishing risk indications at regular intervals. This synchronous REST client, scheduled every 30 s in the experimentation, facilitates continuous information capture by the Andon system in the enterprise layer.
- The data captured from the experiment are stored in two databases: SQL for information traceability and business, and NoSQL for Big Data DB ingestion, providing a historical and analytical data source.

At the enterprise tier, we will have the most visual part of the architecture:

- The frontend for the proof-of-concept uses the angular single-page framework with the Node.js engine, enabling real-time updates and implementing the Andon system for monitoring.
- A simulated client system is implemented using Grafana 10 [63], connected to databases, to display a dashboard with key values. The dashboard triggers a review when the bottleneck alarm is activated, offering insights into data behavior. The platform allows interval-based updates, displaying trends over different periods. Additionally, the enterprise tier will feature an interface to connect to the NoSQL database with historical data from the Big Data system to analyze future trends and make decisions regarding the sensible dataset for bottleneck detection.

The following diagram represented in Figure 12 shows a data flow diagram (DFD) with the interaction between the components of the IIoT architecture experimentation under a software engineering perspective.

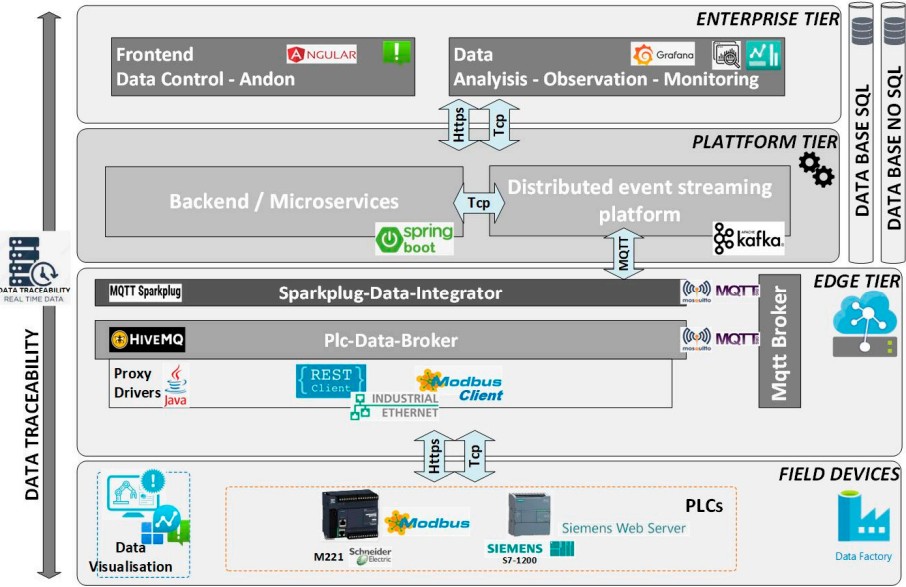

**Figure 11.** Implementation and technological components used in the proposed framework for experimentation.

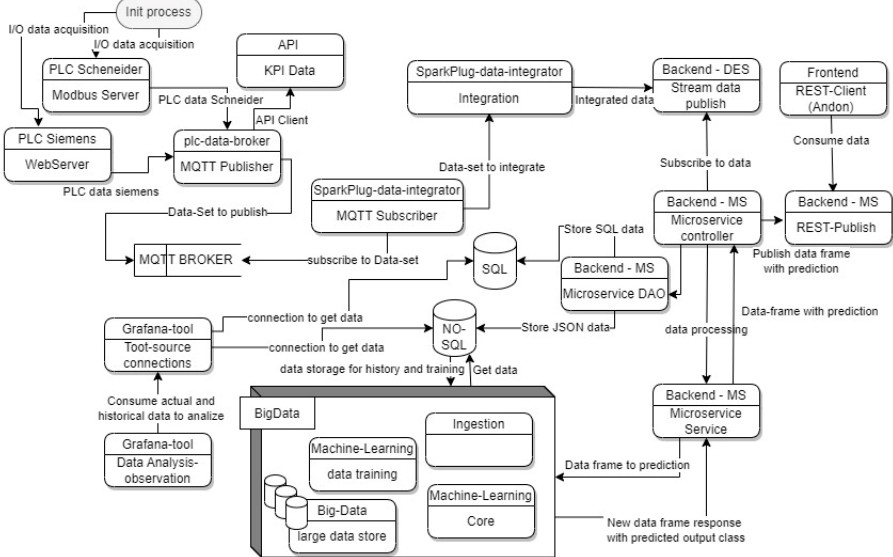

**Figure 12.** Data flow diagram (DFD) of the different software components in the IIoT architecture implemented for experimentation.

## 4.3. Big Data Architecture

The proposed model will be offered by the Spark ecosystem with streams [52]. This tool makes it possible to connect different types of data for subsequent processing. Through the platform tier of the IIoT proposal, connection by event transmission and the NoSQL database, where we have been storing information, will be allowed.

Following the diagram shown in Figure 13, our requirements for the Big Data system in the context of bottleneck detection in manufacturing lines are:

- First, by inserting the set of data detected and integrated from the IIoT architecture into the Big Data system through the NoSQL database, we will obtain a data history. To do this, a data frame will be created with an input with this sensitive dataset for bottleneck detection and an output class to classify it. The Big Data system will train this data framework to generate predictions about the input of new information. An optimized Hadoop distributed file system (HDFS) will be used to preserve historical data over time.
- Second, the distributed event streaming platform will provide real-time data, establishing a connection with Big Data to generate predictions based on the new data and considering the trained historical data. This forecast will indicate whether there is a risk of bottleneck in the Andon system.

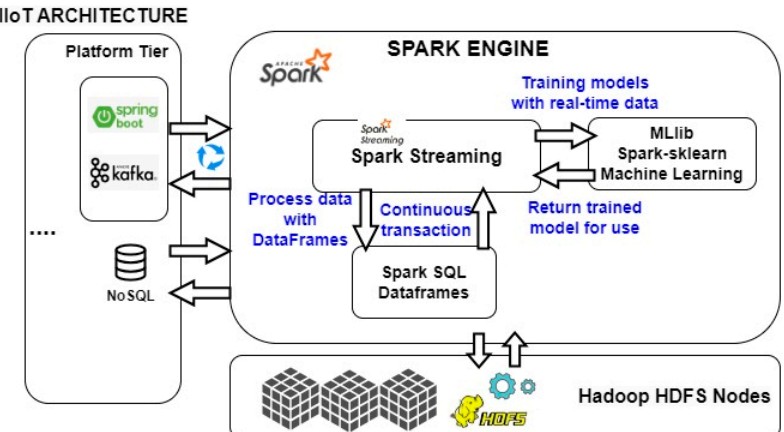

**Figure 13.** Big Data and Machine Learning sequence. Details of integration with the proposed IIoT architecture.

## 4.4. Results

In Figure 14, a simulation displays the obtained results. The Andon system in the frontend exhibits the prediction result based on the data training. This frontend connects to the backend to retrieve the prediction based on the integration of newly detected data with the information trained in Big Data, indicating, if it exists, the presence of a bottleneck. Additionally, the dataset variables were analyzed using an observation and monitoring tool to assess their trends and identify any exceedance of the predefined limits.

Figure 14 displays temperature control and counter engine trends, utilizing dashed lines to demarcate upper, lower, and moderate risk limits. In temperature control, three limits are simulated, while the counter engine graph features upper and moderate risk limits. The green solid line represents actual measurements, highlighting instances where risk limits are exceeded. The counter engine graph includes a vertical blue dashed line, a key marker indicating anomalous measurements. These marked points and lines, such as the blue dashed vertical trace, can be saved for analysis, facilitating comparison with other previously saved points, for example on historical ones to identify bottleneck trends and improve understanding of system performance.

In the experimentation, we utilized the Grafana tool to incorporate the analytical layer, assessing the traceability behavior of the system. Through a comparison graph, we

examined the time taken for information to arrive via each of the protocols employed. The analysis revealed that communication through the Modbus protocol with the Schneider PLC source introduced greater latency compared to communication via the HTTP Web Service with the Siemens PLC source.

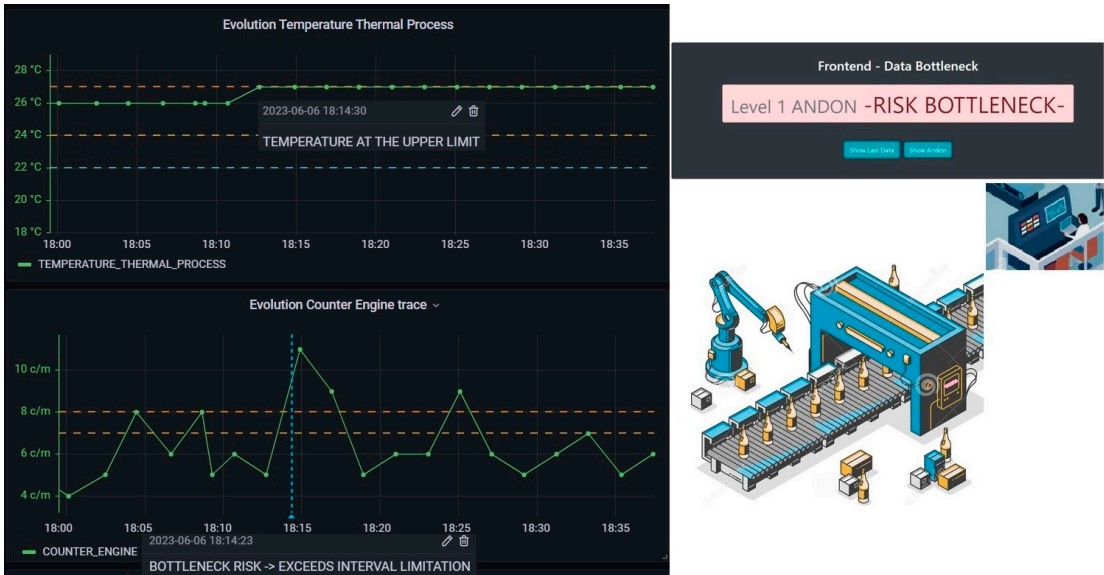

**Figure 14.** Analysis and comparison of detection trends and limits. Simulation of the Andon visual system.

The analysis of the results enabled us to identify points with the most significant delays and review timestamps across different tiers of the proposed architecture. The objective was to identify areas with the highest latency to diagnose network issues, server overload, and communication problems originating from the PLC protocol and interactions among the implemented components in the IIoT architecture.

The blue dashed line in each of the daily graphs on Figure 15 shows the time limit that was set to analyze the trend in the measurements and check for possible latencies.

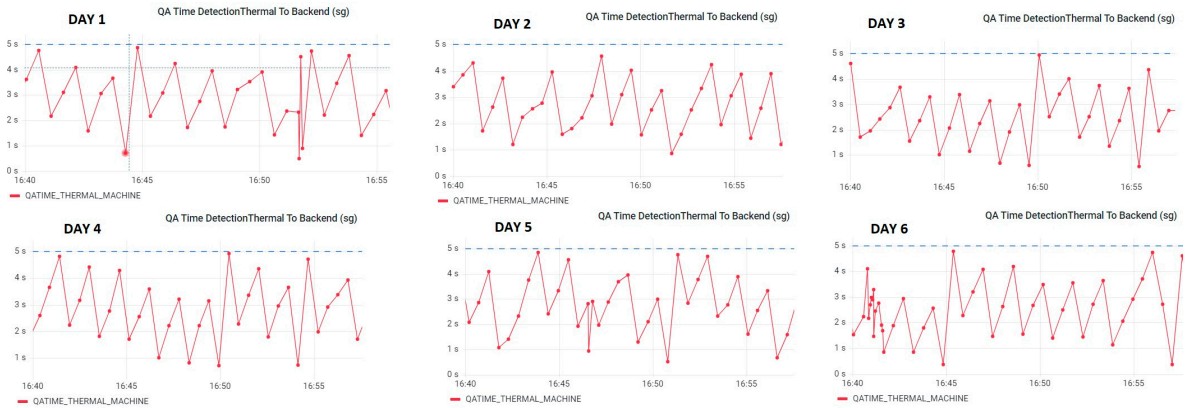

**Figure 15.** Daily chart to analyze trends. This allows us to see the data evolution over time.

In Appendix B, we show the Machine Learning experimentation and results.

The graph shown in Figure 16 alongside Table 1 related to it shows a comparison between the response times of temperature sensors (blue solid line) and counting sensors, respectively (red solid line). In the experimentation, we established a time limit (dashed line) during optimization with trained data that allowed us to overlay different measurements and measure them on a common target. In this case, we used 2 sensor readings; however,

the observability tool will enable crossing a greater number of measurements, displaying, or hiding them as parameterized to monitor risks.

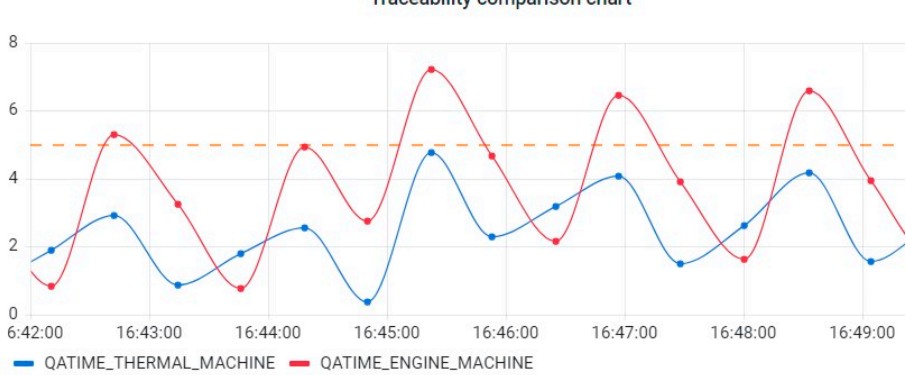

**Figure 16.** The comparison graph (and the related table (Table 1)) shows the evolution of response times in the temperature detection machine in contrast with the motor counting machine. It shows the points at which limits were established during optimization with trained data.

**Table 1.** Comparative table of data with the times captured during the measurement.

| Register_Date | Qa_Time_Engine_Machine (Sg) | Qa_Time_Thermal_Machine (Sg) |
|---|---|---|
| 13 November 2023, 16:42:10 | 1.89 | 0.847 |
| 13 November 2023, 16:42:10 | 1.89 | 0.847 |
| 13 November 2023, 16:42:42 | 5.33 | 2.95 |
| 13 November 2023, 16:43:14 | 3.26 | 0.875 |
| 13 November 2023, 16:43:46 | 0.769 | 1.81 |
| 13 November 2023, 16:45:22 | 7.25 | 4.80 |
| 13 November 2023, 16:46:57 | 6.49 | 4.08 |

## 5. Discussion

### 5.1. Predictive Maintenance as a Reference

Various authors have conducted studies on industrial maintenance, with a focus on predictive manufacturing in smart environments, Industry 4.0 references, and challenges with the Internet of Things [64]. They have examined current paradigms, such as Big Data [10,11,51], Machine Learning [26–29,65], and the broader use of artificial intelligence to develop mechanisms for predicting problems and, ultimately, enhancing and optimizing industrial maintenance. This exploration has enabled the application of the different techniques, tools, and architectures discussed in this manuscript, particularly referencing Architecture 4.0 [13,33,35], and supervisory algorithms utilized in predictive maintenance [6,8,27], adapted for real-time monitoring and ensuring industrial data quality.

In addition to the help that can be provided by the analysis of historical data related to field elements, the experience of the maintenance technician must be considered. The role they have and their knowledge about the machinery is fundamental to be able to perform follow-ups based on experience and serve as a basis for the knowledge of the predictions. In addition, in this context, the technician can determine key variables with the help of information that can impact the bottleneck analysis.

### 5.2. Use of Edge Architecture vs. Cloud Computing in the Proposed Framework

In our proposal, we integrated an edge layer to address the primary goal of having a device layer data with minimal latency, treating them as immediate data, and considering only minimal framework layer latency for bottleneck prediction. This layer operates near the manufacturing environment, providing the advantage of rapid and accurate availability. Dedicated secure edge gateways, as mentioned in [66], are offered by some vendors.

In contrast to cloud computing, sending data to the cloud can introduce delays, and there is the challenge of network unavailability, as highlighted by various authors [67,68]. Therefore, ensuring high data availability and, depending on criticality, employing a different network to mitigate network failures would be essential.

Another aspect to consider is the security level. According to the authors of [68], one of the biggest problems of the Cloud is the data stored with different providers and in different locations. In addition, industrial data transmitted to the cloud are sensitive, as they contain relevant manufacturing information that can be subject to interception and lead to losses [67].

Conversely, in edge computing, we handle and process a minor set of data compared to the extensive processing capability of the cloud. In this context, the authors of [67] proposed diverse architectures to address the limitations of both edge and cloud computing. They also introduced the concept of Fog computing, emphasizing the advantage of establishing a network with lower latency and reduced data load toward the cloud.

Ultimately, the integration of technologies, such as edge, fog, and cloud, aims to foster collaboration among them, with the choice depending on the specific application and its requirements. For our proposed architecture, incorporating a cloud layer at the enterprise level is conceivable. Cloud services could handle analysis, observation, and monitoring, leveraging historical data, while also accommodating newly integrated real-time backend information for relevant comparisons. The front-end component of the Andon system would also necessitate consideration, as it relies on real-time data responses.

### 5.3. Digital Twin

In our proposal, digital twins were not explicitly included; however, there is a potential synergy with external digital twins through the edge layer, enabling real-time interaction. The edge tier acts as a gateway, acquiring information from control devices and smart sensors. The proposed system focuses on integrating data for bottleneck detection, sourced from sensors, actuators in PLCs (as demonstrated in the Siemens and Schneider PLC experimentation), and other IoT devices, such as autonomous sensors. The communication aspects of the proposal involved the use of advanced technologies, including industrial Ethernet, fieldbus, and wireless options, such as 5G or Bluetooth. The edge tier, based on Edge computing, ensures the protection of physical integration heterogeneity, accommodates diverse communication protocols, and provides security and privacy measures [67].

Within the context of integration with the digital twin, physical resources, such as sensors and actuators, will send real-time data to the digital model. The Figure 17 shows a graphical example of integration from the proposed system to the digital twin.

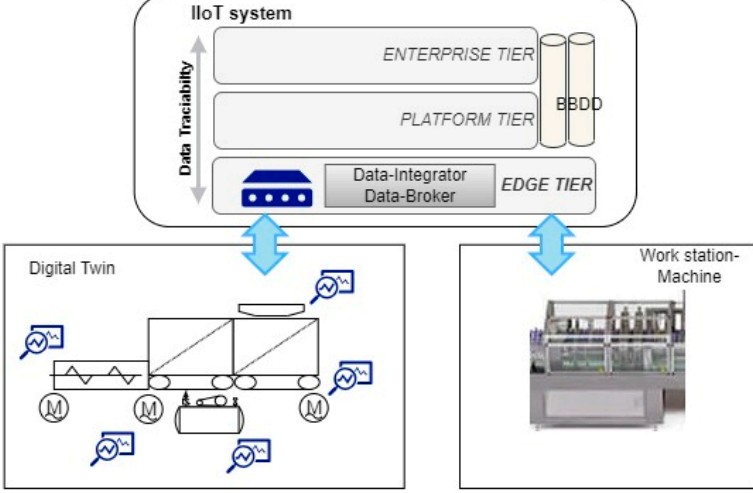

**Figure 17.** Graphical example of integration of the proposed system in the digital twin.

The digital twin offers the opportunity to leverage historical production data. An additional benefit is the real-time control and readjustment of sensors and actuators based on the digital model simulation, enabling optimization without disrupting the actual production line. This capability is particularly valuable for addressing indicators of bottleneck issues, such as throughput.

The integration of the proposed IIoT system with the digital twin can be achieved using the various communication protocols discussed in this manuscript. To ensure proper operation, the digital twin must be precisely aligned with the real machine, ensuring that measurements sent by the integration are correctly synchronized with other components. Challenges in achieving synchronization, particularly in IoT networks with entities with variable resources, are discussed in [69]. These challenges include synchronizing hierarchical component structures and addressing broadcast-response delays, especially in cloud environments.

A possible future line of inquiry could be the reuse of the edge tier to operate on a total transmission of the real machine to a digital twin of its own, which would generate an added value by improving predictions by being able to make changes without damaging the real machine.

Regarding an incremental design, one way will be to follow an incremental approach, according to the typology in [70], from twin components to achieve a twin system that is equivalent to the production line.

### 5.4. Metaverse and Hyperautomation

Azure has introduced a recent initiative in the context of the metaverse, detailed in [71]. This initiative involves the use of Azure Digital Twins tools and Azure Data Explorer to review data history. The focus is on automating updates for networks representing components of a digital twin model. This includes tracing twin lifecycle events and relationship events among multiple digital twins. The initiative aims to create a metaverse environment, enabling the execution of queries between twins for reasoning, using execution tools and time series data. The concept extends to incorporating operational panels for augmented and mixed reality experiences. Practical examples include querying data and monitoring technicians' work status, processes, and resources.

In our analysis, one use case would be to use digital twins to simulate identical production lines in the metaverse. This would allow interactive comparisons during bottleneck situations, analyzing the values plotted at those times to make informed decisions in the future.

The current concept of "hyper-automation", as described in [56], involves automating extensively within an organization using tools such as AI, robotic process automation (RPA), and process mining. The objective is to create a more agile model by eliminating obsolete and repetitive work processes. Hyper-automation goes beyond traditional task automation, emphasizing the maximization of both process and task automation. In a specific application mentioned in [38], hyper-automation is employed for air quality assessment, addressing a data-driven, automated decision-making problem.

### 6. Challenges, Open Issues, and Future Work

In delving into the analysis, several gaps, challenges, and unresolved aspects have become apparent, directly impacting the proposed architecture in the realm of Industry 4.0:

- The challenge lies in outdated Programmable Logic Controllers (PLCs) with non-adaptable communication protocols, hindering Industry 4.0 principles of flexibility, scalability, and seamless integration. This limitation not only affects compatibility but also introduces communication latency with modern systems. Adapting to today's integrable systems becomes a significant challenge for companies, impacting their competitiveness and the smooth transition to Industry 4.0 models.
- Another notable challenge is the computational complexity associated with data processing in a local industrial environment. This challenge affects the feasibility of leveraging available data, particularly when considering access. As highlighted in [10],

many Machine Learning techniques face scalability and generation capacity issues due to the complexity introduced by incorporating Big Data in the cloud. This limitation poses an open issue, necessitating a thorough investigation into the analysis and classification of key quality data to mitigate the processing volume. It emphasizes the importance of focusing on high-quality data relevant to the specific process or problem under analysis.

- In our research focused on Machine Learning algorithms for manufacturing, particularly in predictive maintenance, SVM and KNN classification algorithms were explored for bottleneck analysis in manufacturing lines. In addition to the considerations, the authors of [27] presented a table illustrating the successful application of these algorithms in predicting failures and errors, including experimentation in IIoT sensor networks. However, an open challenge, as mentioned in [28], is the need for improved accuracy in these algorithms, requiring support from suitable feature selection methods. This underscores the difficulty in many cases to identify the key data essential for the required analysis.

In Appendix C, detailed information about the development environments employed for experimentation has been provided.

Here, we acknowledge the limitations and outline future work. We have highlighted the need for a more in-depth study of a specific manufacturing line to thoroughly evaluate the proposed solution. Future efforts involve advancing experimentation by implementing the study on a real manufacturing line, transitioning from a proof-of-concept approach. The focus will shift toward practical application, testing, and measurements within the scope of a specific production line. The ultimate goal is to initiate a project that implements all layers of the proposed IIoT architecture.

Secondly, the research established that the SVM and KNN algorithms proved to be the most efficient for analyzing the bottleneck problem in manufacturing lines, drawing on analogous studies from existing literature in predictive maintenance. To validate their effectiveness, there is a need for measurement in a real manufacturing system. Additionally, ongoing training of information and optimization of the algorithms are recommended to ensure accurate predictions through continual contrast with real manufacturing data.

This research sought to optimize the proposed IIoT architecture by studying existing models in Industrial Big Data and merging data from diverse sources, such as similar manufacturing lines. Furthermore, there are plans to expand experimentation with additional protocols, such as OPC-UA, and explore applications beyond, including testing the quality of the manufacturing process. Additionally, further studies will investigate future considerations introduced by Industry 5.0 with the 5.0 society concept [72], assessing their impact on 6G, which is anticipated to fully integrate artificial intelligence [73].

## 7. Conclusions

The convergence of IIoT and the Industry 4.0 paradigm facilitates the development of a new generation of components, sensors, and actuators, gradually integrating them into manufacturing lines. This ongoing integration of communication protocols and smart systems into operational technology addresses the historical gap that existed with IT technologies. These advancements enhance accessibility to data, enabling seamless analysis and monitoring in both real time and deferred time.

The study of bottlenecks in manufacturing lines has been extensive, often relying on analysis and simulation. This proposal aligns more closely with predictive models and references from Industry 4.0, drawing parallels with disciplines such as predictive maintenance.

This manuscript presents a framework for advanced bottleneck detection in manufacturing lines. The proposed solution involves designing an IIoT architecture with four integrated layers, incorporating a Machine Learning system as a predictive engine to support data analytics for Industry 4.0 systems. Technological advancements in services, communications, and real-time event transmission have made smart sensors essential for providing quality information. These sensors can be integrated into data frames to analyze

complex issues, as discussed in this article. The abundance of data, whether from sensors or indicators, poses challenges related to effective bottleneck detection.

The framework, built on the IIoT architecture and Industry 4.0 principles, offers an integrated solution to the problem at hand. It encompasses phases for capturing information from sensors and actuators, integrating datasets that provide bottleneck information, and facilitating real-time data distribution for predictive modeling with Big Data. Additionally, it enables monitoring and stores historical information for continuous data training. In the experiment, the four layers of the architecture were operational, executing various phases of the designed process. The selected sensors were categorized and prepared for integration, forming the data frame. Considering their real-time acquisition, the platform for event transmission persistently provided information along with timestamps for distribution.

The architecture provided us with a set of classified data in real time, persisted in its efforts, generated historical data, and predicted the bottleneck risk through a trained Machine Learning model provided by Big Data. In addition, it monitored and analyzed its traceability to detect latencies. As a result, we have obtained an improved detection of bottlenecks in production lines and a predictive indicator to help make decisions in the business context and in manufacturing management.

**Supplementary Materials:** The following supporting information can be downloaded at: https://www.mdpi.com/article/10.3390/app14010323/s1.

**Author Contributions:** Conceptualization, M.J.R.A.; methodology, M.J.R.A.; validation, M.J.R.A. and I.A.C.; proposal, M.J.R.A.; writing—original draft preparation, M.J.R.A.; writing—review and editing, M.J.R.A.; conceived and designed the experiments, M.J.R.A.; reviewing, I.A.C.; supervision, I.A.C. and J.A.C.S. All authors have read and agreed to the published version of the manuscript.

**Funding:** This work was supported in part by the Spanish Ministry of Science, Innovation, and Universities, under Project PID2022-142043NB-I00.

**Institutional Review Board Statement:** Not applicable.

**Informed Consent Statement:** Not applicable.

**Data Availability Statement:** The data presented in this study are available in Supplementary Materials

**Conflicts of Interest:** The authors declare no conflicts of interest.

## Appendix A. Basic Flow of the Data Preprocessing

We started from the knowledge of the connectivity of the field devices within the domain of the manufacturing line. Control elements, such as the PLC, were connected to the detectors and actuators that were going to be used for the analysis of their measurement, and in addition, to consider autonomous sensors where the problem is analyzed. This connectivity will be provided by different protocols, such as Modbus, Ethernet, or wirelessly connected elements.

Once we have the key sensitive bottleneck dataset selected for analysis, the connection to the IIoT system will be established, with the source as the PLC, associated machine or station, etc.

The security level was set both at the application level on the data-broker component of the edge tier and in the field elements. For example, for HTTP connectivity with a PLC server, security will be added at the SSL level through certificates, both at the source and the destination.

As for the data to be acquired from the PLC, we prepare the measurements to be sent to the IIoT system (if they were not programmed previously). Figure A2 illustrates a program fragment implemented in a Siemens S7-1200 PLC with the transformation of an analog temperature signal ready to be sent to the IIoT architecture.

In the case of data that do not come from the industrial controllers, such as key production indicators: throughput, cumulative production data, delivery times, etc., the

data-broker component application will be provided with the integration of these. For example, with a REST API.

The application was executed at the edge tier with the heterogeneous dataset and its connections to the field elements. It publishes the data over the MQTT broker as Topics, together with the traceability data consisting of the timestamp. As an example, two code snippets of an implementation written with java at the edge tier are shown in Figure A3.

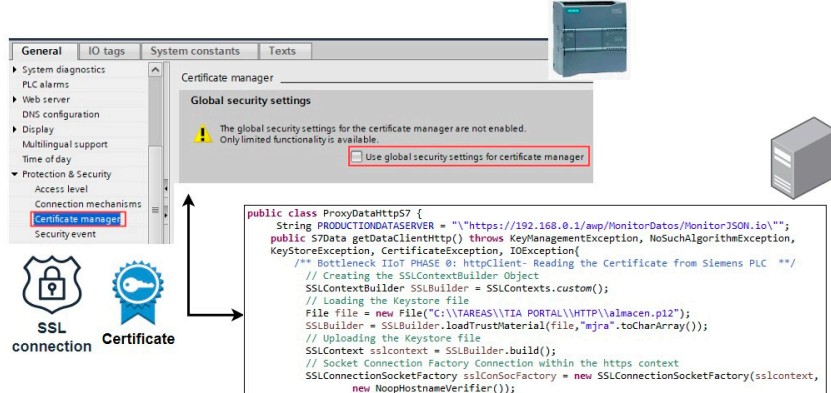

**Figure A1.** Https security configuration to link data between the PLC and the edge data-broker component.

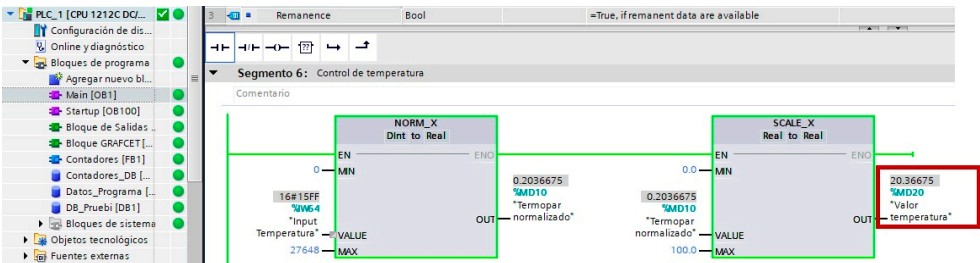

**Figure A2.** Program fragment implemented in a Siemens S7-1200 PLC with the transformation of an analog temperature signal.

```java
// Send MQTT topic temperature conveyor engine PLC Telemecanique M221
client.toBlocking().publishWith()
        .topic("Devices_line_1/NDATA/edgeNode_1/temperature_engine")
        .payload(getTemperature())
        .send();
TimeUnit.MILLISECONDS.sleep(500);
// Send MQTT topic temperature thermal process PLC Siemens S7-1200
client.toBlocking().publishWith()
        .topic("Devices_line_1/NDATA/edgeNode_2/temperature_thermalProcess")
        .payload(getTemperatureThermalProcess())
        .send();
/**
 * Return from Telemecanique M221 PLC Conveyor engine
 * with modbus protocol temperature motor key value
 * @return
 */
private static byte[] getTemperature() {
// simulate a temperature sensor with values between 20°C and 30°C
//      final int temperature = ThreadLocalRandom.current().nextInt(20, 30);
    DataModelPLC data = modbusClientM221();
    final String temperature = data.getTemperatureEngine() + ";" + data.getTimestampM221();
    return ((temperature)).toString().getBytes(StandardCharsets.UTF_8);
}
/**
 * Return from Siemens S7-1200 PLC ThermalProcess machine with
 * Web Server https protocol temperature thermal process
 * @return
 */
private static byte[] getTemperatureThermalProcess() {
    // simulate a temperature sensor with values between 20°C and 30°C
    //      final int temperature = ThreadLocalRandom.current().nextInt(20, 30);
    S7Data s7data = siemensPLCClientHttp();
    final String temperature = s7data.getTemperatura1() + ";" + s7data.getRegisterDate().getTime();
    return (temperature).toString().getBytes(StandardCharsets.UTF_8);
}
```

**Figure A3.** Code fragment with data acquisition from PLCs and publication on the MQTT broker.

From here, the data will be published in the MQTT broker and, therefore, will be ready for processing within the IIoT architecture.

### Appendix B. Machine Learning Experimentation and Results

From the proposed Big Data architecture, we have experimented with the Machine Learning component. During the experimentation, we created an algorithm to train the dataset of the detection performed through the IIoT system, simulating the Machine Learning part of Big Data. The objective was to obtain an optimized and trained model to have the best coverage. To do this, we experimented with a dataset with 10,000 records after having the system up and running for the past few months.

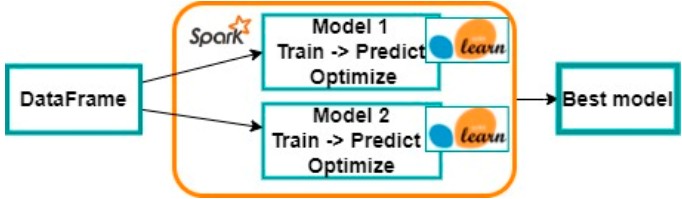

**Figure A4.** Machine Learning structure for optimization of the trained model.

We created the dataset with the experienced sensors as input. The sequence we followed was to start the designed IIoT system and process the data on the different layers. Once in the platform layer, the distributed event streaming component Kafka [61] will be consumed in the backend, providing the information in real time and storing it continuously on the designed databases. In addition, the time trace was stored for analysis. An example of a log in JSON format is shown in Figure A5.

[{"idBn":11486,"temperatureEngine":"25","timestampTemperatureEngineM221":1700055505461,"counterEngine":"6","timestampCounterEngineM221":1700055504932,"processStepEngine":"1","temperatureThermalProcess":"23.69792","timestampTemperatureThermalPS71200":1700055507380,"thermalProcessStep":"1","timestampSparkPlug":1700055507717,"timestampKafkaDataDistribution":1700055507720,"timestampBackend":1700055508474,"detailExecutionData":"getbottleneckdata","registerDate":"2023-11-15T13:38:28.482+00:00","version":1,"qaTimeEngineMachine":3.542,"qaTimeThermalMachine":1.094}]

**Figure A5.** Integrated data in JSON format.

We imported the chosen input dataset and the output class of the data frame across 10,000 records into a data frame according to the following snippet and output class with the resulting combination of bottleneck indicators: indicator_1 OK and indicator_2 RISK.

**Table A1.** Data frame model to train information.

| Temperature_Engine | Temperature_Thermal_Process | Process_Step_Engine | Machine_Counter_Engine | Class |
|:---:|:---:|:---:|:---:|:---:|
| 19 | 21 | 8 | 5 | indicator_1 |
| 19 | 20 | 1 | 4 | indicator_2 |
| 25 | 26 | 3 | 6 | indicator_1 |
| 26 | 20 | 1 | 8 | indicator_2 |
| 20 | 24 | 5 | 6 | indicator_1 |

We created a classifier for the Support Vector Machine (SVM) and k-Nearest Neighbors (KNN) classification algorithms. We trained them with the records, subsequently performing the prediction with a new dataset of 20 records. We then optimized both algorithms by introducing features such as cross-validation. This method divides the dataset into k-folds and compares the data in one-fold to a model that has been trained with data in the other k-folds. After evaluating all the information, it is possible to evaluate the overall performance of the model. For the experimentation, we used 5-fold cross validation.

Finally, we saw how the models improved toward 0.988 for KNN and, specifically, 0.999 for SVM. Thus, the SVM algorithm demonstrated the best coverage. We obtained the

following result, shown in Figure A7, provided in the implemented phase "predict output results" for the optimized SVM.

**Result prediction with k-nearest neighbors (KNN)**

```
In: y_prediction = knn.predict(X_new)
    print("Prediccion: {}".format(y_prediction))

Prediccion: ['indicator_1' 'indicator_1' 'indicator_1' 'indicator_1' 'indicator_1'
 'indicator_1' 'indicator_1' 'indicator_1' 'indicator_1' 'indicator_1'
 'indicator_2' 'indicator_2' 'indicator_1' 'indicator_2' 'indicator_2'
 'indicator_1' 'indicator_2' 'indicator_2' 'indicator_2' 'indicator_2'
 'indicator_2' 'indicator_2' 'indicator_2' 'indicator_2']
```

**Result prediction with support vector machine (SVM)**

```
In: y_prediction = svc_cv.predict(X_new)
    print("Prediccion: {}".format(y_prediction))

Prediccion: ['indicator_1' 'indicator_1' 'indicator_1' 'indicator_1' 'indicator_1'
 'indicator_1' 'indicator_1' 'indicator_1' 'indicator_1' 'indicator_1'
 'indicator_2' 'indicator_2' 'indicator_2' 'indicator_2' 'indicator_2'
 'indicator_2' 'indicator_2' 'indicator_2' 'indicator_2' 'indicator_2'
 'indicator_2' 'indicator_2' 'indicator_2']
```

**Figure A6.** Prediction results in experimentation with the KNN and SVM algorithms.

**Optimize model SVM**

```
In: import numpy as np
    from sklearn.model_selection import GridSearchCV

    Cs = [0.001, 0.01, 0.1, 1, 10]
    gammas = [0.001, 0.01, 0.1, 1]
    param_grid = {'C': Cs, 'gamma' : gammas}
    svc_cv = GridSearchCV(svc, param_grid, cv=5)
    svc_cv.fit(X, y)
    svc_cv.best_params_
    svc_cv.best_score_

Out: 0.9994781474233528
```

**Predict Output results from new input data**

```
In: df_new = pd.read_excel(r'C:\Users\mrodr\Documents\machine_learning\caso_bottleneck\machine_le
    X_new = df_new.values

In: y_prediction = svc_cv.predict(X_new)
    print("Prediccion: {}".format(y_prediction))

Prediccion: ['indicator_1' 'indicator_1' 'indicator_1' 'indicator_1' 'indicator_1'
 'indicator_1' 'indicator_1' 'indicator_1' 'indicator_1' 'indicator_1'
 'indicator_2' 'indicator_2' 'indicator_2' 'indicator_2' 'indicator_2'
 'indicator_2' 'indicator_2' 'indicator_2' 'indicator_2' 'indicator_2'
 'indicator_2' 'indicator_2' 'indicator_2']
```

**Figure A7.** SVM model optimization parameters and prediction results in the new data framework.

The Jupyter Lab development environment [74], widely used in the industry for data analysis as a lab subsystem for data analytics and Big Data, was used as the technology to carry out this comparison. In this Lab, we performed the workflow to train the information and optimize and perform comparisons between algorithms within the context of data science and Machine Learning, continuously. For this, we used the scikit-learn [75] libraries, widely used for the purpose of data analytics in Big Data systems. Once the algorithms were implemented and tested, we integrated them through the spark-sklearn library within the Spark framework for Big Data. The IIoT system via the backend will connect to the service offered by Spark with the trained information. Each newly acquired dataset was run over the algorithm to predict if there is a bottleneck risk.

## Appendix C. Development Environments Employed for Experimentation of the Proposed IIoT Architecture

To develop industrial operating components at the device level with Siemens and Schneider PLCs, the environments provided by the manufacturer were used. On the one hand, TIA Portal version 16 [76] was used, which has support for the web server and OPC-UA for Siemens. On the other hand, the EcoStruxure Machine Expert laboratory environment [77] for Schneider was used. To implement components in the edge and platform tiers, Eclipse IDE [78] was used. For the enterprise tier, we used visual studio code for the Andon system development and the Grafana system with its own environment. In addition, we used tools such as Maven, Git, or Jenkins to automate it and integrate it into a CI/CD system [79], which will allow us to scale, evolve, and add work for future challenges more easily in the context of bottleneck detection in manufacturing lines in an Industry 4.0 environment.

In summary, to implement the experimentation, we used up-to-date technologies and protocols from Industry 4.0, such as Modbus in Ethernet, MQTT and HTTPS with SSL, architectural focuses, such as distributed stream processing and microservices, and software languages, such as Java, Python, and JS.

The experimentation with Big Data was carried out in the development environment with Spark and the laboratory environment for data science with Jupyter [74]. The latter

will provide us with an environment to optimize the algorithms and experiment with future challenges.

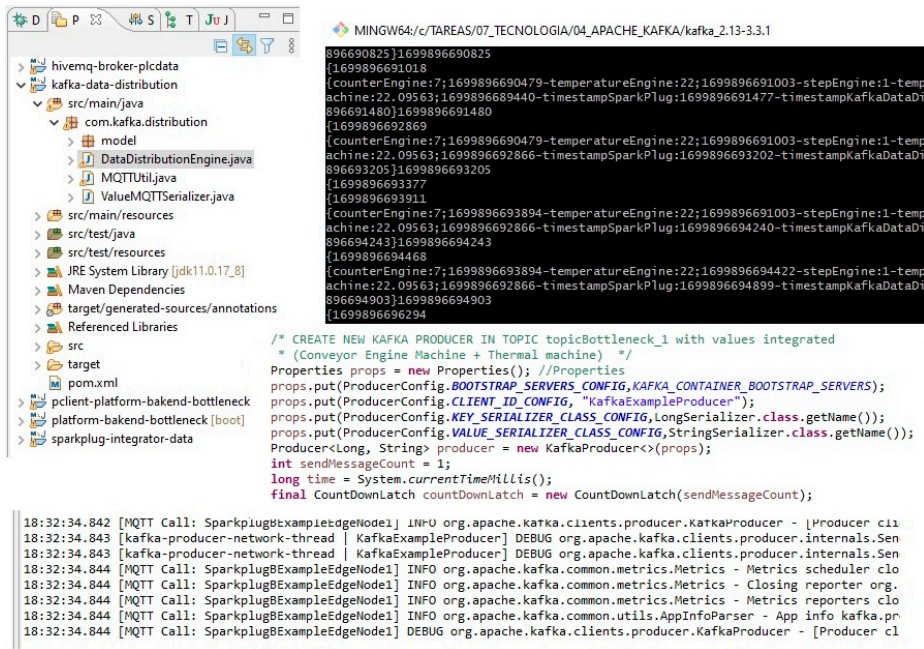

**Figure A8.** Snapshot with the project's structure, and Kafka execution console with data frames and console logs during experimentation execution.

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
