# Peer review of "IIoT System for Intelligent Detection of Bottleneck in Manufacturing Lines"

_applsci, doi:10.3390/app14010323_

Round 1

Reviewer 1 Report

Comments and Suggestions for Authors

The authors have primarily devised a operational framework grounded in the existing smart factory, the IIoT, the Industry 4.0 paradigm, and big data. They propose a system using a dataset associated with the problem to demonstrate the risk and indicators for intelligent bottleneck analysis and detection. To realize this, the authors have defined an architecture with multiple levels for real-time data acquisition. Moreover, the proposal facilitates the generation of historical information used for training and continuous optimization. In summary, the paper is well-motivated and addresses publishable issues, but certain aspects need further enhancement:

1. Clarify the connection between presented notions such as IIoT and cyber-physical systems. Discuss additional concepts like digital twin, metaverse, and hyperautomation (e.g., 10.1007/s12559-022-10101-8).

2. Specify the novelty or study motive of this paper compared to other mature working frameworks based on the current smart factory in Section 1.

3. Include references to recent review papers and new concepts, such as 10.1016/j.ijcce.2023.06.001.

4. Improve the quality of some figures, particularly Figure 1, by using clearer images.

5. In Section 3, provide explanations on how to process multi-modal information and perform pre-data processing.

6. In Section 4, consider presenting Data Flow Diagram (DFD) modeling figures from the perspective of software engineering.

7. Address study limitations in the last section. Clarify the relationship between HDFS and SPARK in Figure 11, as they appear to have similar functions.

Author Response

Dear reviewer,

Thank you very much for the points made about the review, I think they are very interesting to take into account in the manuscript.

I will give you a summary of what I have added on the points you indicated:

On the point they made to clarify the connection between notions presented as IIoT and cyber-physical systems, digital twin, metaverse and hyperautomation, I have added a new discussion section (5). In this one, I have extended my research into cyber-physical aspects such as the digital twin, the metaverse, and hyperautomation. These are detailed in sections 5.3 and 5.4 respectively.

Regarding the item Specify the novelty (GAP) or reason for study of this work compared to other mature frameworks based on the current smart factory in Section 1, content has been introduced in this section.

Included the references they needed and improved the quality of the images. In the case of the previous Figure 1 it has been removed due to a copyright issue.

Explanations on how to process multi-modal information and perform pre-data processing have also been provided in Section 3, in point 3.2 following Figure 6.

A Data Flow Diagram (DFD) figure of the components of the proposal experienced from the software engineering point of view has also been included. In this sense it has not been possible to extend it due to lack of time to other diagrams. It can be seen at the end of subsection 4.2.

The relationship between HDFS and SPARK has been clarified in subsection 4.3.

Also, regarding the references, new ones have been added from 66 to 80.

Finally, limitations are discussed in section 6 before the conclusions.

Best regards

Manuel J. Rodriguez

Reviewer 2 Report

Comments and Suggestions for Authors

In general, this is an interesting paper since it reports on a very complex project within the field of manufacturing. It is more or less a case study that includes both the setup of an IIoT environment and a report on experiences with it. I don't see too much novelty in terms of architectural design etc.

Still, I am optimistic that the paper can be published but from my point of view, there are a few major requirements that should be addressed in a revision:

- What you describe has a lot to do with predictive maintenance where, in the same way, sensor data is used to predict when machines might fail. I suggest that you take this topic into account and look into solutions already proposed in the literature.

- There will be more reports available on such a system as you describe it, and I suggest looking for these - there will be implementations based on one of the two IIoT architectures you mention.  

- To me, it is not clear how you did the step from Sections 1/2 to Section 3. In Chapter 1, you describe your desire to design a system that can predict bottlenecks, but the topic "bottleneck" is not covered in Section 2. This leads to the question, why you believe that the architecture you design in Section 3 should be a solution to the bottleneck problem. In addition, it remains completely unclear whether your architecture has any relationship with the two IIoT architectures presented in Section 2. It should be made clear whether your architecture is an implementation of one of the two, and if not, why not (and why, then, you presented these architectures).

Figures 7 and 11, by the wy, are identical.

Comments on the Quality of English Language

The paper is readable, but the quality of the language should be improved. It would definitely help to have someone read it who is a native English speaker. The text is too much influenced by the original language.

Author Response

Dear reviewer,

Thank you very much for the points made about the review, I think they are very interesting to take into account in the manuscript.

I will give you a summary of what I have added on the points you indicated:

About the research in the maintenance area, I included at the time in the introduction different aspects related and that have some analogy or are also assimilated with the bottleneck problematic. Specifically on predictive maintenance I also show different references in the chapter on big data, for example 11 and 26 to 28. Now in a new discussion section that I have added at the request of the reviewers I have put a subsection of predictive maintenance as a reference for this study, the detail can be seen in subsection 5.1. In the conclusion I also add some point about this aspect.  

On the point that indicated the existence of reports on similar systems like the one I raise (based on the reference architectures of industry 4.0) I have added a paragraph in point 2.4. on articles where I have seen similar systems which are based on the reference architectures. In this you can see the detail of the point.

About the bottleneck, although I had the research part done since the manuscript is based on the proposal to deal with this problem, I had not put a specific point, since I explained it in part in the introduction. Now I have added point 2.1 in relation.

I also clarify on which reference the IIoT proposal is based, this I add at the end of point 2.4.

The previous figures 7 and 11 as you indicated were identical. There was a confusion in inserting them. Now with the changes you have requested me they are identified as 10 and 16.

I also inform you that at the request of other reviewers I have added point 5 of discussion.

Also, regarding the references, new ones have been added from 66 to 80.

Best regards

Manuel J. Rodríguez

Reviewer 3 Report

Comments and Suggestions for Authors

Overall the presented content is well-structured and novel. However, I have the following comments for the authors to consider

1. It will be better to have a dedicated discussion about edge vs cloud computing architecture with respect to the proposed framework. Consider the aspects of computing latency and security.

2. One of the most relevant aspects of machine learning is the explainability and interpretability aspects. Comment on how it is integrated/planned as part of the proposed framework.

3. How similar or different is the proposed IIoT system from a digital twin for manufacturing lines? If different, how does it integrate with an existing digital twin system in terms of communication and security standards?

Author Response

Dear reviewer,

Thank you very much for the points made about the review, I think they are very interesting to consider in the manuscript.

I will give you a summary of what I have added on the points you indicated:

On the first point, I added a new discussion section (5). In this I have incorporated the review on the use of edge architecture with respect to cloud computing, the detail you will see in subsection 5.2.

On the second point, which concerned the aspects of explicability and interpretability, I have also added a subsection 3.3.3. Where I expand the manuscript on this aspect. Also, in this section you can see the detail.

The last point you have made, also very interesting about digital twins, I have inserted in subsection 5.3 of the new discussion section. There you will see the details I have inserted.

Also, regarding the references, new ones have been added from 66 to 80.

Best regards

Manuel J. Rodríguez

Round 2

Reviewer 1 Report

Comments and Suggestions for Authors

Accept in present form

Author Response

Dear reviewer,

Thank you for the review. Just to inform you that, at the request of one of the reviewers, we have condensed the paragraphs without removing content, taking into account the points you had requested in the previous review. Also, at the reviewer's request, we have moved the more technical points to appendices.

Best regards

Manuel J. Rodriguez

Reviewer 2 Report

Comments and Suggestions for Authors

While the authors have addressed my main comments from Review #1, it has not led to the paper being better readable. The overall content is okay - as far as it is possible to identify it within the vast amount of information provided. I have two major suggestions which would help to make the paper a good one:

1.) It is absolutely necessary to work on the language - and I do NOT (only) mean the English language, I mean the annoying and disturbing use of wring vocabulary and connotations. There is not something like "intelligent information"and you cannot "prevent the performance from working efficiently". There are so many of these issues that it is impossible to list them all. You need to work together with IT experts who will help you improve the use of IT vocabulary, because it is simply not possible to understand what you mean.

2.) The paper is too long, with the effect that the "red thread" is not visible. There are far too many details on the technical solution, with hundreds of different single technologies which no one can keep in mind. There are far too many repetitions - The new Section 2.1 is nearly a complete repetition of the introduction. I strongly suggest strictly streamlining the paper and maybe putting the technical descriptions in an annex or an online addendum or similar.

Comments on the Quality of English Language

In addition to the things said above, please have the paper read by a native English speaker - or even by one of the AI language improvement tools. It is worth it ....

Author Response

Dear reviewer,

Regarding the points that have been included for review:

We have thoroughly reviewed the vocabulary and rephrased certain paragraphs to enhance clarity, eliminating any incorrect semantics. We have done this to the best of our ability, considering that without specific data, it can be more challenging to determine what to remove and what to keep.

In the course of the review, we have trimmed paragraphs and removed repetitive elements that we observed. We have reworked section 2.1 and moved more technical parts to various appendices. However, due to the document's context, we cannot entirely eliminate technical terms as they are part of the explanation of the points.

We have also taken into account the need for a native English speaker to review these changes.

Best regards

Manuel J. Rodríguez